# Preinitiation Complex Loading onto mRNAs with Long versus Short 5′ TLs

**DOI:** 10.3390/ijms232113369

**Published:** 2022-11-02

**Authors:** Benjamin Weiss, Pascale Jaquier-Gubler, Joseph Alphonsus Curran

**Affiliations:** 1Department of Microbiology and Molecular Medicine, University of Geneva Medical School, 1211 Geneva, Switzerland; 2Institute of Genetics and Genomics of Geneva (iGE3), University of Geneva, 1205 Geneva, Switzerland

**Keywords:** translation initiation, eIF4E, 5′ TL

## Abstract

The first step in translation initiation consists in the recruitment of the small ribosome onto the mRNA. This preinitiation complex (PIC) loads via interactions with eIF4F that has assembled on the 5′ cap. It then scans the 5′ TL (transcript leader) to locate a start site. The molecular architecture of the PIC-mRNA complex over the cap is beginning to be resolved. As part of this, we have been examining the role of the 5′ TL length. We observed in vivo initiation events on AUG codons positioned within 3 nts of the 5′ cap and robust initiation in vitro at start sites immediately downstream of the 5′ end. Ribosomal toe-printing confirmed the positioning of these codons within the P site, indicating that the ribosome reads from the +1 position. To explore differences in the eIF4E-5′ cap interaction in the context of long versus short TL, we followed the fate of the eIF4E-cap interaction using a novel solid phase in vitro expression assay. We observed that ribosome recruitment onto a short TL disrupts the eIF4E-cap contact releasing all the mRNA from the solid phase, whereas with a long the mRNA distributes between both phases. These results are discussed in the context of current recruitment models.

## 1. Introduction

Translation is the most energy consuming process in the cell [1]. It consists of four successive steps (initiation, elongation, termination and subunit recycling) but it appears to be regulated mainly, although not exclusively, at the initiation phase. During initiation, the 40S ribosome subunit assembles with the eukaryotic initiation factors (eIF) eIF1, eIF1A, eIF3, eIF5 and the ternary complex (TC) eIF2.GTP.tRNAiMet to make the 43S pre initiation complex (PIC) (Figure 1A). According to current models, the PIC generally loads onto the mRNA 5′ end through interactions between its eIF3 subunit and eIF4G with the former being positioned at the mRNA exit channel on the ribosome [2,3] (Figure 1A). The latter is a scaffolding protein that forms part of the eIF4F complex that sits on the 5′ cap structure. It is a trimolecular complex composed of the cap binding protein eIF4E, the RNA helicase eIF4A and eIF4G (Figure 1A) [4]. The eIF4E-cap interaction is critical for the ultimate positioning of the PIC at the 5′ end of eukaryotic mRNAs. Interactions between the subunits of eIF4F serve to stabilize the complex on the cap. The DEAD-box helicase activity of eIF4A acts to unwind secondary structural elements downstream of the cap creating a “landing site” for the PIC. This is stimulated by binding to eIF4G and by the accessory proteins eIF4H and eIF4B (Figure 1A). Following loading, the PIC scans the 5′ transcript leader (TL) in a 5′ → 3′ fashion. When an AUG codon enters the ribosomal P-site, it triggers a series of events that releases the eIFs and promoting recruitment of the 60S subunit. The now active 80S ribosome can then enter the elongation phase [3].

Therefore, AUG recognition is the crucial event that marks the end of the initiation phase and the entry into elongation. The efficiency of start codon recognition is influenced by the sequences that flank the AUG. When these are sub-optimal, 43S ribosomes can sometimes bypass the AUG codon without initiating and then start at downstream sites, a process referred to as leaky scanning. In mammals, the Kozak context was identified as the optimal sequence for efficient start codon recognition. This motif consists of a −3 purine and a +4 guanine relative to the A of the AUG [4] and was found to be conserved in mRNAs from many vertebrates [5]. Furthermore, it was observed that even non-AUG codons embedded in a good Kozak context could initiate translation, albeit very inefficiently [6]. Context is read by the eIF1 and eIF1A factors that are respectively positioned close to the P and A-sites on the ribosome (Figure 1A) [7,8]. The eIF1 actually stabilizes an open conformation of the mRNA channel on the ribosome that facilitates mRNA attachment and sliding during scanning [9,10]. Sequence context is thought to stabilize the ribosome and facilitate base pairing between the tRNAiMet and the start codon. This, in-turn, triggers the release of the inorganic phosphate generated by eIF2.GTP hydrolysis in the TC, an event stimulated by eIF5. The subsequent ejection of eIF1 promotes the closed conformation that effectively clamps the ribosome down on the mRNA over the start codon [11].

Despite being positioned in a good Kozak context, AUG codons proximal to the 5′ end are poorly recognized and are generally very leaky [12]. As the 43S ribosome protects an mRNA footprint of 30 nt, 17 nt of which are upstream of the P-site [13,14], translation from AUG codons 20 nt downstream of the 5′ cap is not efficient and are generally neglected when predicting the protein readout of a transcript. This type of expression analysis is further complicated by the intrinsic 5′ transcriptional start site (TSS) heterogeneity associated with mammalian Pol II promoters [15,16,17]. Nonetheless, studies identified a sequence motif that was found to promote translation, and impair leakiness, from an AUG codon closer than 20 nt to the 5′ cap [18]. This motif was named TISU (Translation Initiation of Short 5′ UTR). Its sequence, SAASATGGCGGC (in which S is C or G), played a role both in the regulation of transcription and translation [19]. Translation from TISU is eIF4A, and therefore 5′-3′ scanning independent, and requires the release of eIF4G and possible the entire eIF4F complex from the cap [18]. Furthermore, initiation on short 5′ TLs carrying a TISU sequence are insensitive to the over-expression of eIF1, a factor that plays an important role in reading codon context [18,20], suggesting mechanistic differences in the reading of the “Kozak context” and the “TISU context”. However, it remains unclear if the mechanism by which the PIC is initially recruited onto a TISU transcript is fundamentally different. With the canonical 5′ cap dependent loading model for the ribosome, one would require a subsequent 3′-5′ back-scanning to allow the AUG codon to be positioned in the P-site. However, it remains unclear if “retro-scanning” is a feature unique to transcripts with short 5′ TLs and, if so, how it is triggered. In an alternative mechanism, recently proposed by the lab of Pestova, the mRNA was postulated to thread over the surface of the ribosome with the AUG entering initially through the A-site [21]. This model readily explains how the ribosomal P-site can read all sequences starting from the +1 position on the mRNA and is essentially independent of the AUG position and hence TL length (Figure 1). 

In this manuscript, we have investigated translation initiation on short 5′ TLs. Using a reporter, we show initiation can occur efficiently in vivo on short 5′ TLs independent of TISU context and was readily detectable on a TL of only three nucleotides. Nonetheless, TISU was found to be the least leaky. In vitro, we observed robust initiation events on TLs as short as 1 nt and employed ribosome toe-printing to validate the position of the 80S over the start site. We have probed the mechanism of PIC recruitment by following the fate of capped mRNAs carrying either a long or a short 5′ TL immobilized on a solid matrix via the eIF4E cap binding protein. Upon the addition of translationally active cell extracts, the behavior of these two mRNAs is noticeably different and these observations are discussed in the context of the current models for ribosome recruitment onto the 5′ cap.

## 2. Results and Discussion

### 2.1. Translation Initiation Can Occur on Very Short 5′ TLs 

According to the in vitro studies reported by Marilyn Kozak, leader sequence less than 20 nt are very inefficient and susceptible to leaky scanning, even if the AUG context is favorable [12]. Nonetheless, initiation events have been reported on AUGs positioned close to the cap and the TISU motif (CAAGATGGCGGC) has been proposed to promote efficient non-leaky initiation on short 5′ TLs [22]. As a starting point for our current study, we re-examined the translation of an mRNA carrying 5′ TLs of various length using our previously described LP^Next^ translation reporter [23]. This reporter bears several AUG codons in overlapping ORFs (all protein products carry a HA epitope tag) and can be used to monitor translation initiation (the first start site is referred to as AUG^ELK^) and leaky scanning (initiation on a series of downstream or internal AUGs referred to as iAUG^a/b/c^: Figure 1B). The proteins expressed from each ORF have similar intracellular stabilities, hence steady state levels reflect initiation site efficiency [24] and since both products arise from the same mRNA template it avoids the need to normalize to transcript levels. Generating mRNAs with a unique and defined 5′ end is critical for the interpretation of the assay. As polymerase II promoters produce heterogeneous transcription start sites (TSS) [17], mRNAs were generated using the T7 bacteriophage polymerase whose TSS is well defined [25]. The LP^Next^ reporter constructs were generated in a plasmid vector carrying only a T7 promoter. We created reporters with 5′ TLs of 1 nt (G), 3 nts (GGG), an 8 nt TL referred to as Rdm (it has a weak TISU sequence), an 8 nt TL carrying a TISU element and a standard TL of 54 nt (derived from the β actin 5′ TL) with a good Kozak context (Figure 1C). Apart from the 1 nt construct, all others started with 3 Gs as this is optimal for T7 transcription [25]. To transiently express in vivo, the plasmids were transfected into HeLa cells infected with a vaccinia virus recombinant expressing T7. This virus not only expresses the polymerase but also provides the cytoplasmic capping activity [26]. In parallel, the same plasmids were used to generate in vitro T7 run-off transcripts that were subsequently capped by the vaccinia virus-capping enzyme and 3′ polyadenylated with a commercial poly(A) polymerase (see Section 4). Capping was monitored by eIF4E binding, and polyadenylation by gel electrophoresis (Appendix A). These capped/polyadenylated RNAs were used to program a rabbit reticulocyte cell lysate (*RRL*). The translational readout from both systems was monitored by immunoblotting using an anti-HA Ab (Figure 1D). In living cells, the G construct gave weak but detectable initiation events at the cap-proximal AUG (AUG^ELK^) with the major translational product arising from the downstream iAUG^a/b/c^ (these products co-migrate) [24]. Remarkably, when this same transcript was expressed in vitro, the main translation product arose from AUG^ELK^. This effective inversion of the profiles between in vivo versus in vitro was even more evident with the GGG construct (Figure 1D). Expression from Rdm, TISU and Kozak all confirmed the leakiness of AUG^ELK^ in the infected cells compared to the RRL (Figure 1D). Whereas TISU and Kozak promoted initiation exclusively at AUG^ELK^ in the RRLs, in viral infected cells these same mRNAs were very leaky, with iAUG^a/b/c^ being the major product detected (see Table 1). Vaccinia virus 5′ TLs are generally short and it has been reported that initiation on short 5′ TLs is promoted during viral infection, which is clearly not what we have observed [27]. However, many viruses, including vaccinia, exploit leaky scanning to increase their decoding complexity, which may explain the leakiness of the AUG^ELK^ (https://viralzone.expasy.org/1976, accessed on the 21 September 2022) [28]. Mammalian (“virus-free”) cell lines constitutively expressing T7 polymerase exist, however, the T7-derived transcripts are not capped and are only efficiently express if they carry a 5′ IRES element, rendering this approach unsuitable for our current study [29].

Robust initiation from the 1 and 3 nt TL was observed in the RRL despite the fact that the codon is very leaky. However, translation can also initiate from non-AUG codons [6]. To confirm that the expression pattern was not due to a non-AUG initiation event downstream of AUG^ELK^, we performed toe-printing in RRLs. Toe-printing consists in the mapping of either the 80S or 43S ribosome on in vitro transcribed mRNA by limited primer extension. We first confirmed the length of the in vitro T7 mRNAs by primer extension (Figure 1E). 

We performed 80S toe printing on the 1 and 3 nt TL transcripts by pre-treating the RRL with cycloheximide (CHX). As a control, the RRLs were supplemented with 10 mM MgOAc that inhibits ribosome loading [30]. Primer extended cDNA products were analyzed on a sequencing gel (Figure 1F). With CHX we observed a multi-band toe-print at +19–23 nts upstream of the AUG^ELK^ with CHX. The same toe-print was observed on the long Kozak mRNA (Appendix A). These results confirm the assembly of an 80S ribosome over the AUG^ELK^ in RRLs on the 1 and 3 nt TLs, a result consistent with the protein expression study (Figure 1D). 

### 2.2. TISU Reduces Leaky Scanning on a Short 5′ TL

Our expression studies in RRLs and vaccinia virus infected HeLa cells revealed minor differences in the expression profiles between Rdm and TISU, specifically concerning AUG^ELK^ leakiness. However, both systems have flaws; the viral context in the DNA transfected living cell promotes leaky scanning and the relative 5′ cap independence of the RRL (Appendix A). To overcome this, we tested the same series of 5′ TL reporters by transfecting their in vitro generated capped/polyadenylated transcripts into HEK293T cells and using the same RNAs to program HeLa cell extracts. Unlike RRLs, expression in HeLa extracts is highly 5′ cap dependent (Appendix A). However, we were unsuccessful in using these extracts for toe-printing analysis probably because of the endogenous RNaseH activity [31]. As observed in the viral infected HeLa cells, expression from the AUG^ELK^ was barely detectable in HEK293T cells transfected with the 1 nt 5′ TL (G) transcript but it was much more robust in the cell extracts (Figure 1G: Table 1). Once again, and in both expression systems, the 3 nt 5′ TL should robust but leaky initiation at AUG^ELK^. Regarding AUG^ELK^ codon efficiency (its non-leakiness), the long 5′ TL with the Kozak context was clearly the most efficient in both expression systems. An analysis of the expression patterns from the two 8 nt 5′ TLs (Rdn versus TISU), indicates that the TISU element context rendered the start codon less leaky in-line with previous reports (Table 1). 

In conclusion, we observed robust, but leaky, initiation on a 5′ TL as short as 3 nt in four different expression systems. We even observed weak in vitro expression from the single nucleotide TL. These results confirm that the PIC can read sequence starting from the beginning of the mRNA transcript, an observation consistent with the results obtained in a reconstituted mammalian expression system [21], and although TISU may favor AUG recognition and reduce leakiness, it is not an essential element. 

### 2.3. Translation from a Short 5′ TL Is Cap Dependent

Our results indicate that the ribosome can start translation efficiently from 5′ TLs as short as 3 nt in vivo and 1 nt in vitro. We examined if translation from these mRNAs was cap dependent. To do so, we constructed a bicistronic mRNA carrying the renilla (RLuc, first cistron) and the firefly (FLuc, second cistron) luciferases (Appendix A). RLuc was positioned downstream of either a short 5′ TL of 8 nt or a longer 5′ TL derived from human β-actin with a good Kozak [32], while FLuc translation was under the control of the EMCV IRES. The mRNA was produced with or without the addition of a 5′ cap and reporter activity was monitored in two in vitro mammalian expression systems (namely, RRL, HeLa). We did not observe major differences in the RLuc first cistron read-out between capped short and Kozak TLs in the RRL and HeLa systems. The RLuc activity was decreased around two-fold in the absence of a 5′ cap group independent of TL length, while FLuc activity was increased (Appendix A). This probably reflects the competitive advantage of the EMCV IRES over the non-capped 5′ [33]. Significantly, RLuc expression was highly cap dependent in the HeLa extracts, consistent with previous reports (Appendix A) [33]. These results highlight the following: RRLs are a poor in vitro system to monitor 5′ cap dependence of translation, an observation already noted by a number of investigators [34].The expression profiles in both systems reveals no evident difference in the behavior of the short and long 5′ TLs with regards to cap-dependence. Translation from the short 5′ TL is consequently cap-dependent.

### 2.4. PIC Loading

If translation from a short 5′ TL requires the cap, we were interested to understand what happens to the eIF4E-mRNA contact as the ribosomal P site positions over the start codon. In the conventional loading model, and on long 5′ TLs, the 43S PIC is recruited onto the mRNA via eIF4F-eIF3 interactions (Figure 1A) and then scans 5′-3′ (Figure 2A). Consequently, there is no de facto prerequisite for the eIF4E-cap interaction to be broken during loading whilst the eIF4E-43S interaction is lost during scanning [35]. However, with a short 5′ TL, positioning of the start codon in the P site would involve a 3′-5′ retrograde movement or realignment (Figure 2A). Back scanning can occur, at least within the context of IRES-mediated ribosome recruitment [36], and a bidirectional walking model for the scanning ribosome has already been proposed [35]. Concerning TISU elements, the TISU-ribosomal protein contacts reported by the Dikstein lab could promote this realignment event post-recruitment [37]. One intriguing possibility is that these same contacts could also induce release of PIC complexes that have not initiated at the start codon thereby limiting leaky scanning on a TISU motif. With short TLs, such scenarios would presumably require an eIF4E-cap dissociation during realignment and prior to 60S recruitment due to steric hindrance (Figure 2A). The release of eIF4G on TISU elements has been observed but the fate of eIF4E remains unclear [38]. Another model proposes that the mRNA threads over the ribosome with the AUG codon first entering via the A site (Figure 2B). In this scenario, the AUG is threaded first through the A-site, with the 5′ end now positioned on the ribosome surface or within the mRNA channel [21]. Steric restrictions would presumably also require at least a transient eIF4E-5′ cap dissociation prior to threading as eIF4E cannot be accommodated in the RNA channel on the ribosome surface [21]. Cross-linking studies suggested that the eIF4E-5′ cap interaction was lost during PIC recruitment in a reconstituted system [21]. This model readily explains how the ribosome P site can read all nucleotides downstream of the 5′ cap. Threading would occur irrespective of 5′ TL length. In an attempt to explore these alternative models, we initially sought to covalently crosslink eIF4E to the 5′ cap of our bicistronic transcripts with the aim of impairing mRNA threading, and then monitor the effect on short versus long 5′ TLs. However, we were unsuccessful. We choose instead to incubate in vitro generated capped mRNAs expressing RLuc with bacterially expressed human ^HIS^eIF4E immobilized on His-Dynabeads (Figure 2C). The mRNA binding was highly efficient and stable independent of 5′ TL length (Figure 2D/E), and in the presence of cell extracts an eIF4F complex appeared to assemble on the bead (Figure 2F). The immobilized ^HIS^eIF4E-capped transcript was used to program an in vitro translation system (Figure 2C). In this configuration, mRNA threading and retro-scanning would be impeded or would require mRNA dissociation from the ^HIS^eIF4E. Reactions were stopped by placing the tubes on ice and the bead (BF: binding fraction) and supernatant (NBF: non-binding fraction) fractions were recovered. Reporter activity in the NBF was recorded, and RNA from both fractions was recovered by Trizol extraction. We originally employed RRLs, but all translation on the solid phase was inhibited independently of the 5′ TL.

Low RRL translation efficiency of mRNA immobilized on solid phase was reported in an earlier study and was attributed to the accumulation of protein aggregates on the bead surface that impeded ribosome recruitment [39]. Considering these problems, and the fact that the RRL is an expression system that is poorly cap-dependent (Appendix A), we decided to re-visit the experimental approach using HeLa cell extracts. After 10 min incubation, reporter activity was recorded and the mRNA in the BF and NBF fractions was monitored by semi-quantitative RT-PCR. The PCR cycles had been optimized to assure that we were within the linear range (Figure 2D). If, as a consequence of 43S recruitment, the eIF4E-mRNA contact is disrupted we should observe that the mRNAs are released from the beads. They should then redistribute between the immobilized and soluble eIF4E pools. The short incubation time limited the rounds of translation but nonetheless allowed us to monitor significant reporter readouts. We also confirmed that during the incubation period no ^HIS^eIF4E protein leeched from the beads into the supernatant (Figure 2F). Curiously, we observed significant differences in the fraction of the mRNA retained on the solid phase when comparing a long and a short TL. With the short construct, the majority of mRNA (~90%) was released, whereas with the long the major fraction remained bead-associated (~70%), despite the fact that both transcripts gave similar reporter (RLuc) readouts (Figure 2E). This pattern was observed in four independent experiments and points to a clear mechanistic difference between the long and short TLs during recruitment (Figure 2E: lower panel). Furthermore, despite multiple attempts we were unable to detect 18S rRNA (using RT-PCR) or the rpS6 (using immunoblot) in the BF at levels significantly above background levels (BSA treated beads), suggesting that very little translation is occurring on the bead surface. 

Despite the relatively short incubation times, the earlier assay probably monitored multiple rounds of initiation. To restrict the assay to a single initiation event we added CHX or GMPPNP to the HeLa extract prior to addition of the beads. The addition of either drug did not markedly alter the behavior of the 5′ TL short: the majority was still released from the bead: (Figure 3A). However, with the long 5′ TL the presence of CHX, and the subsequent stalling of the 80S ribosome over the AUG^ELK^, we observed a significant release of the mRNA from the bead effectively inversing the phenotype observed initially (Figure 3B). We suspected that the queuing of scanning 43S ribosomes upstream of the paused 80S could trigger this event (Figure 3C). Such a phenomena has been evoked to explain non-cognate initiation on an AUU codon within the 5′ TL of the AZIN1 mRNA [40]. To test this, we replaced CHX with GMPPNP and asked if this queuing would occur upstream of a paused 43S. Like CHX, the drug severely inhibited RLuc reporter expression but, unlike CHX, it did not trigger a release of the 5′ TL long transcript from the bead (Figure 3B). This would lead us to propose that in the presence of GMPPNP the pausing of the 43S over the start codon is disrupted by the upstream scanning PICs, an event that relieves ribosome queuing (Figure 3C). Independently of the exact mechanistic model of PIC recruitment, these studies highlight a difference in the behavior of mRNAs carrying either a long or a short 5′ TL.

Concerning 5′ cap accessibility and eIF4E binding, there is one clear difference between the 5′ TL long and short. Independent of the route by which the PIC is initially recruited (loading versus threading), what assembles immediately downstream of the 5′ cap in a short TL is an 80S ribosome whereas this is not the case with a long 5′ TL, in which it is the 43S that scans away from the 5′ end. Assuming that the eIF4E-5′ cap interaction is broken at each loading event, it means that the subsequent re-binding of eIF4E may be different in the context of a downstream 80S (short TL) versus a scanning 43S (long TL). With regards to the loading and threading models of PIC recruitment, it cannot be excluded that both mechanisms are occurring but that only threading permits productive initiation events on a short TL. Additionally, it is possible that the environment created by the presence of elongating 80S ribosomes immediately downstream of the 5′ cap on a short TL actually promotes initiation by threading. It could be envisaged that only the initial recruitment onto a 5′ TL short is eIF4E dependent with the subsequent 43S threading events being eIF4E independent. Since it is likely that no active translation occurs on the solid matrix this could explain the behavior of the short TL subsequent to its release from the bead. The distribution of the long TL between BF and NBF may reflect its use of both recruitment modes.

### 2.5. Future Directions

The 5′ TL is clearly a central element in the regulation of the translational readout and consequently the cellular proteome. One key feature is its length, as defined as the distance in nucleotides between the 5′ cap and the first AUG codon. It is now accepted, and confirmed in the first part of our current manuscript, that start codons within the first twenty nucleotides can be recognized by the PIC. This in-turn raises mechanistic questions about how the ribosome assemblies on the 5′ cap, in particular concerning the eIF4E-cap interaction. Our solid phase assays hints at differences in PIC recycling that correlate with TL length. However, the assay is limited by the fact that only two TL lengths were anaylzed and a more extensive length analysis is clearly required if a solid correlation is to be established. Furthermore, it reveals no new insights into the fate of the cap-bound eIF4E during PIC recruitment and subsequent start site selection. Our attempts to covalently couple eIF4E to the 5′ cap failed. However, other avenues could be explored to enhance the eIF4E-cap interaction and ask if this differentially influences initiation from short versus long 5′ TL reporters. This could include modified cap structures with lower Kds for eIF4E [41] or cap-biotinylated RNAs that can be subsequently UV cross-linked to eIF4E [42]. The mode of PIC recruitment and the fate of eIF4E during loading onto short versus long 5′ TLs remains elusive.

## 3. Materials and Methods

Cell culture, infection, and transfection: Experiments were performed in HeLa cells. The cells were cultured in Dulbecco’s Modified Eagle Medium (DMEM: Gibco, Paisley, UK) supplemented by 5% fœtal bovine serum (Gibco, Paisley, UK) and 1% penicillin/streptomycin (Gibco, Paisley, UK). They were grown at 37 °C in a humidified 5% CO_2_ chamber. When 50–60% confluent, cells were infected with a vaccinia virus recombinant expressing T7 polymerase at a multiplicity of infection >4 for 2 h in DMEM serum free. The media was then replaced with fresh growth media. A few hours later, the cells were transfected using the FuGENE^®^ HD Transfection Reagent (Promega, Madison, WI, USA) according to the manufacturer’s instructions. Cells were collected after an overnight incubation and lysed in RIPA buffer (150 mM NaCl, 10 mM Tris HCl pH 7.8, 1% deoxycholate, 1% Triton X-100, 0.1% SDS). 

Transfections of HEK293T cells were performed using Lipofectamine 3000 (Invitrogen, Waltham, MA, USA) when the cells were 70–80% confluent. Eight hours post-transfection, the medium was replaced with normal growth medium, and lysates were prepared at 24 h post-transfection.

DNA construction: The original LP^Next^ clone was previously described [23]. The ORF starting at the ELK^AUG^ was cloned downstream of the T7 promoter and a series of different 5′ TLs (both in length and sequence) were generated using the primers listed below. All constructions were built in the pGL3 basic backbone and cloned SmaI/XbaI.

Forward

G: 5′TAATACGACTCACTATAGATGGACCCATCTGTGGGG: 5′TAATACGACTCACTATAGGGATGACTTCGAAAGTTTATGARdm:5′CGCTAGCTAATACGACTCACTATAGGGAGAAAATGACTTCGAAAGTTTATTISU: 5′-TAATACGACTCACTATAGGGACAAGATGGCGGCATCTGTGACGCTGTGGReverse: 5′-TCAGCGAGCTCTAGCATTTAGGTG

In vitro transcription and translation: pGL3 constructs were linearized with XbaI and RNA was transcribed using the mMessage mMachine™ T7 Transcription Kit (Invitrogen) and purified by LiCl precipitation according to the manufacturer’s instructions. Transcripts were capped using the vaccinia capping system (NEB) and polyadenyated using the Poly(A) tailing kit (Invitrogen), according to the supplier’s protocol. The 3′ tailing was monitored using semi-denaturing agarose gel electrophoresis. Briefly, RNAs were suspended in formamide loading buffer (95% formamide, 10 mM EDTA, 0.05% bromophenol blue, 0.05% Xylene cyanol), heated at 90° for 2 min and rapidly chilled on ice before loading onto a native agarose gel. Gels were stained with ethidium bromide. 

250 ng of in vitro prepared mRNA was translated in a 50% nuclease treated rabbit reticulocyte cell lysate (RRL, Promega) for 1 h at 30 °C in a final volume of 20 µL. The reaction mix was diluted in an equal volume of RIPA buffer plus Laemmli buffer (250 mM Tris HCl pH 6.8, 40% glycerol, 20% β-mercapto-ethanol, 10% SDS, 0.05% Bromophenol blue) and 10 µL was resolved on a SDS-polyacrylamide gel. 

Immunoblot: 20 µg of protein cell lysate prepared in Laemmli buffer was resolved on SDS-polyacrylamide gel and electro-transferred to PVDF. Antibodies used in this study were anti-HA (Covance clone 16B12) and goat anti-mouse HRP secondary antibody (Bio-Rad, Hercules, CA, USA). 

Toe-printing: Toe-printing was performed as previously described [43]. Briefly, 20 pmol of primer P1 (5′TGCATCCACCAGCTTGAATT) were 5′ labelled with 100 µCi of [γ-32P] ATP (3000 Ci/mmol) using 20 U of T4 DNA kinase (Thermofisher, Waltham, MA, USA) at 37 °C for 1 h. 7 µL of RRL supplemented with 40 U of RiboLock (Thermofisher) was incubated for 5 min at 30 °C with 2 µL of 10 mg/mL cycloheximide (CHX) (Sigma, St. Louis, MO, USA) for mapping the 80S ribosome. 300 ng of mRNA in 2 µL of H2O was denatured at 95 °C for 5 min and cooled rapidly to room temperature before being added to the treated RRL. The mix was incubated for 10 min at 30 °C. 8 µL of a mix composed of 125 mM Tris-HCl (pH 8.3 at 25 °C), 187.5 mM KCl, 7.5 mM MgCl_2_, 25 mM DTT, 2 pmol of ^32^P labelled primer, 0.6 mM dNTP and 100 U of M-MLV reverse transcriptase (Promega) was then added and the incubation continued for 1 h at 30 °C. The RT reaction was terminated by the addition of 20 µL H_2_O and 40 µL phenol/chloroform. Glycogen (10 µg) was added to the aqueous phase before being precipitated with NH_4_OAc/ethanol. The pellet was resuspended in 5 µL of formamide dye (95% formamide, 0.05% bromophenol blue, 0.05% xylene cyanol FF) and 2 µL was loaded onto an 8% polyacrylamide-urea gel alongside a sequencing ladder (Jena Biosciences, Jena, Germany).

eIF4E purification: The N-terminally His tagged human eIF4E was expressed in the *E. coli* strain BL21αDE3 using the pET30 expression vector (kindly supplied by Dr Franck Martin, Strasbourg, France). Transformed bacteria were grown at 37 °C until an A600 of 0.7 and expression induced by the addition of 0.4 mM IPTG (isopropyl-1-thio-β-D-galactopyranoside). Bacteria were incubated overnight at 16 °C and them pelleted and washed in cold PBS. Cells were resuspended in lysis buffer (20 mM HEPES pH 7, 100 mM KCl, 1 mM EDTA, 2 mM DTT, 10% (*v*/*v*) glycerol) using 1/40th of the initial culture volume. Cells membranes were broken by sonicating four times for 30 s on ice. The insoluble inclusion body fraction was recovered by pelleting at 30,000× *g* for 30 min. It was washed and re-pelleted three times in a minimum of 10 mLs of wash buffer (1M guanidine HCl, 20 mM HEPES, 2 mM DTT, 10% (*v*/*v*) glycerol) before solubilizing in 6M guanidine HCl, 50 mM HEPES, 2mM DTT and 10% (*v*/*v*) glycerol (using 1/40th of the initial culture volume). Resuspension was aided by sonication. Insoluble debris was pelleted by centrifugation as 43,000× *g* for 30 min. The soluble protein fraction was then diluted to a concentration <100 μg/mL in solubilization buffer before dialysis/renaturation against 50 mM HEPES, 2mM DTT and 10% (*v*/*v*) glycerol using Spectra Por MWCO 3500 membranes. The cap binding protein was first purified on Immobilized γAminophenyl-m7GTP (C10-spacer) (Jena Bioscience). Binding was performed in dialysis buffer and eluted in 2M KCl, 50 mM HEPES, 0.5 mM EDTA, 2 mM DTT and 10% (*v*/*v*) glycerol. Protein was dialyzed against 100 mM KCl, 50 mM HEPES and 10% (*v*/*v*) glycerol and then adjusted to 1mM β-mercaptoethanol. The final purification was performed on a TALON Metal Affinity Resin (Clontech) using buffers that contained 1mM β-mercaptoethanol. Protein quality was assayed by SDS PAGE and was confirmed RNase-free by incubating with cellular rRNA at 37 °C. 

Preparation of translationally active mammalian cell lysates. Extracts were prepared following the protocol outlined in Terenin and coworkers [44]. Actively dividing cells (≈70% confluence) from a 100 mm petri dish were scrapped into ice-cold PBS (final volume 1 mL) and recovered by pelleting at 1000× *g* for 5 min at 4 °C. They were washed a second time with PBS before resuspending in 200 μL of Lysolecitin lysis buffer (20 mM HEPES-KOH pH 7.4, 100 mM KOAc, 20 mM DTT, 0.1 mg/mL (*w*/*v*) Lysolecitin) for precisely 1 min on ice before centrifuging at 10,000× *g* for 10 secs at 4 °C. The pellet was resuspended in an equal volume of ice-cold hypotonic extraction buffer (20 mM HEPES-KOH pH 7.5, 10 mM KOAc, 1 mM MgAc, 4 mM DTT containing complete protease inhibitor cocktail-EDTA minus). Cells were disrupted in a pre-cooled Dounce homogeniser using 20–25 strokes, transferred to an Eppendorf tube, and spun at 10,000× *g* for 10 min at 4 °C. The supernatant was collected and aliquoted. For translation studies, these cell extracts worked best when fresh. They could be stored at −80 °C for short periods but freezing-thawing reduced activity.

RNAs (100 ng) were translated in a 50% cell extract supplemented with an energy mix buffer (10X: 150 mM HEPES, 80mM creatine phosphate, 200 μg/mL creatine kinase, 10 mM ATP, 10 mM GTP, 200 μM amino acid mix, 5 mM spermidine, 200 μg/mL calf liver tRNA, 10 mM MgOAc, 10mM DTT) for 2 h at 30 °C in a final volume of 10 μL. The reactions were stopped by the addition of Laemmli SB buffer (X4 buffer: 250 mM Tris HCl pH 6.8, 40% glycerol, 20% β-mercaptoethanol, 10% SDS, 0.05% Bromophenol blue.

Translation in the solid phase: beads from 50 μL of Dynabeads His-Tag slurry were recovered and washed twice with 1 mL of Triton Binding Buffer (TBB: 150 mM KCl, 10 mM HEPES pH7, 0.2% (*v*/*v*) Triton X100). They were then resuspended in 1 mL of TBB containing 20 µg/mL yeast tRNA and divided in two equal fractions. Into one faction was added 4 μg of purified ^HIS^eIF4E. The second was incubate with 4 μg of BSA. Both were incubated with gentle agitation for 60 min at 4 °C. They were then recovered and washed twice with TBB before being each further divided into two equal fractions. Each bead fraction was recovered and resuspended in 500 µL of TBB-yeast tRNA plus 2 µg of capped mRNA possessing either a long or short TISU 5′ TL. Beads were incubated a further 90 min at 4 °C with gentle agitation. They were then washed twice with TBB and once with Binding Buffer (BB: 150 mM KCl, 20 mM HEPES pH7, 1.5 mM MgOAc) and then recovered as the free beads. These were resuspended in 5 μL mammalian cell translation mix (see above) on ice before transferring to 30 °C for 10 min with regular agitation. Beads were recovered and washed with TBB. Reporter activity was measured in 1 μL of the supernatant and RNA was isolated from the bead fraction (Binding Fraction) and the rest of the supernatant (Non-Binding Fraction) using Trizol. The quantity of reporter RNA in each fraction was determined by RT-PCR using the Qiagen one step kit with the following primers: (-VE): GAACACCACGGTAGGCTGCGAAATG(+VE) GATCAAAGCAATAGTTCACGCTGAAAGTGTAG

When testing the effect of CHX/GMPPNP the beads were incubated in the same translation mix as above but containing either 1 μg/μL CHX or 2 mM GMPPNP. The mix was incubated 10 min at 30 °C and then processed as above. 

## 4. Conclusions

Using a range of in vitro and in vivo systems, we have demonstrated efficient initiation on AUGs positioned only 3 nts downstream of the 5′ cap. Using immobilized ^HIS^eIF4E as a reporter of 5′ cap accessibility, we have demonstrated clear differences in the behavior of transcripts carrying either a long or short TL suggesting mechanistic differences in their mode of 43S loading.

## Figures and Tables

**Figure 1 ijms-23-13369-f001:**
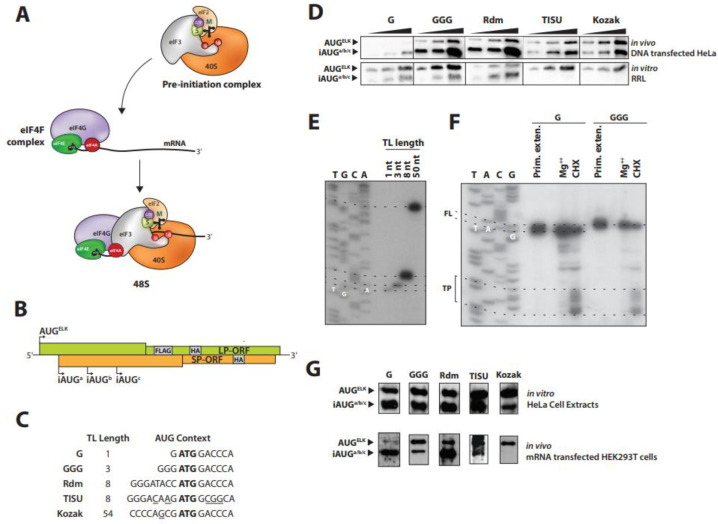
Translation can start on short TLs in vitro and in vivo. (**A**) Conventional model for PIC assembly and loading onto the mRNA 5′ cap via eIF4F. The eIF4E-cap binding is visible as is the key contact between eIF4G and eIF3 on the PIC. The helicase activity of eIF4A unfolds RNA downstream of the cap creating a “landing site” for the 43S. The TC (eIF2.GTP.tRNAiMet) is positioned in the ribosomal P site. The location of the other initiation factors is also indicated. (**B**) Schematic representation of the LP^Next^ translation reporter. The LP^Next^ was cloned downstream of the TL under the control of a T7 promoter. Translation starts at the first start codon (AUG^ELK^), and if there is leaky scanning at internal out-of-frame downstream AUGs (iAUG^a/b/c^). Both ORFs carry a common HA tag. (**C**) The sequences flanking the AUG^ELK^ codon in the clones that were generated. Underlined are indicated the key sequences in both the TISU and Kozak context. (**D**) Anti-HA immunoblots showing the expression profile from LP^Next^ reporters in HeLa cells infected with vaccinia T7 and in RRLs. Because of the high variation in the expression levels from the AUG^ELK^ and iAUG^a/b/c^ in the different constructs, we loaded varying amounts of extract (5, 10 and 20 µg of protein). This assisted in the quantification of the individual bands (see Table 1). (**E**) Primer extension analysis on the +1 (G) and +3 (GGG) transcripts. On the left side is a sequencing ladder. (**F**) Toe-prints observed on mRNAs with +1 (G) and +3 (GGG) 5′ TLs. The position of the 80S ribosomes on the mRNA was monitored by treating the RRL with cycloheximide (CHX). Primer extension was performed using a 32P labelled primer. Primer extension (Primer) was also performed on free RNA to determine the position of the full-length (FL) cDNA product. Addition of high Mg2+ to the RRL prior to addition of mRNA served as a negative control. The cDNAs were loaded onto an 8% sequencing gel alongside a sequencing ladder. The position of the ATG codon is indicated on the ladder as are the ribosome induced stops (TP). (**G**) Anti-HA immunoblots from the LP^Next^ reporters carrying the TLs as indicated. Upper panel: In vitro expressed capped/polyadenyated mRNAs transfected into HEK293T cells. Lower panel: The same mRNAs used to program HeLa cell extracts.

**Figure 2 ijms-23-13369-f002:**
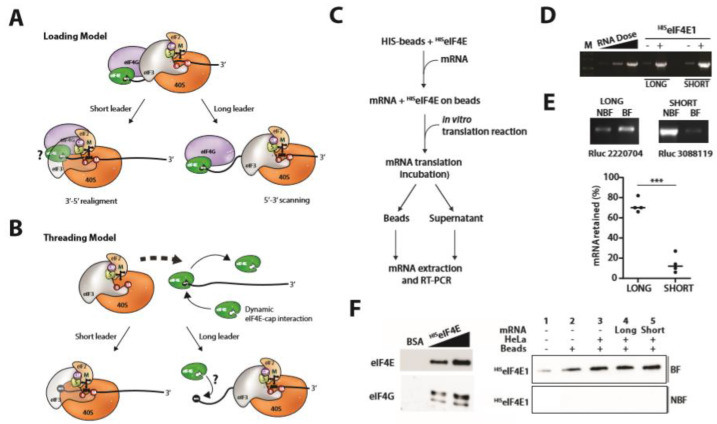
Translation from short and long TLs immobilized on a solid phase matrix via ^His^eIF4E: fate of the transcript. Schematic representations of 43S recruitment via the loading (**A**) and threading (**B**) models on long versus short 5′ TLs. The different initiation factors are all indicated. The fate of eIF4E during both processes is indicated. Loading onto a 5′ TL long does not necessarily oblige eIF4E release since no realignment post-recruitment is required, however, this is unclear with a 5′ TL short (indicated as an ?). In the threading model the cap enters the 43S ribosome via the A site and this presumably requires, due to steric constraints, eIF4E release as the transcript enters the RNA channel on the surface of the ribosome. This would occur independent of TL length. (**C**) Schematic representation of the experimental protocol. HIS-Dynabeads carrying bacterially expressed ^HIS^eIF4E were used to immobilise in vitro generated capped mRNAs carrying either a short (8 nt) or a long Kozak TL fused to an RLuc reporter gene. As a control, beads were incubated with the same amount of BSA. These were then incubated in a cell extract for 10 min. Bead/binding fraction (BF) and supernatant/non-binding fractions (NBF) were separated and the RNA extracted (an aliquot of the supernatant was retained to measure RLuc reporter activity). Transcript levels in both fractions was measured by RT-PCR. (**D**) Agarose gel analysis of the RT-PCR amplicon. Transcripts carrying either a capped long or short 5′ TL were incubated with HIS-Dynabeads carrying (+) or not carrying (-) ^HIS^eIF4E. Bound RNAs were recovered and analyzed by RT-PCR. M indicates size markers. An RNA dose is included (1 ng, 10 ng and 50 ng of an RLuc transcript) to show that the signals are in the linear range. (**E**) A representative RT-PCR analysis of the NBF and BF fractions after incubation in cell extracts. The measured RLuc activity in the cell extract (arbitrary units) is indicated below each panel. Bands were quantitated using the BIO-RAD image-lab software. The experiment was repeated four times and the results are indicated in the lower panel as the % of RNA retained on the bead (BF/(BF + NBF) × 100). The horizontal bar indicates the mean from these experiments. (**F**) Left hand panel: One µg and 2 µg of ^His^eIF4E was immobilized on HIS-Dynabeads. As a negative control, the beads were incubated with 2 µg of BSA. They were then added to a cell extract, recovered, and washed twice (see Section 3) before adding protein sample buffer. Proteins were analyzed on immunoblots using ant-eIF4E and anti-eIF4G Abs. Right hand panel: Translation reactions were performed as above. NBF and BF fractions were recovered and the distribution of the ^HIS^eIF4E protein was monitored by immunoblot. *** *p* < 0.001.

**Figure 3 ijms-23-13369-f003:**
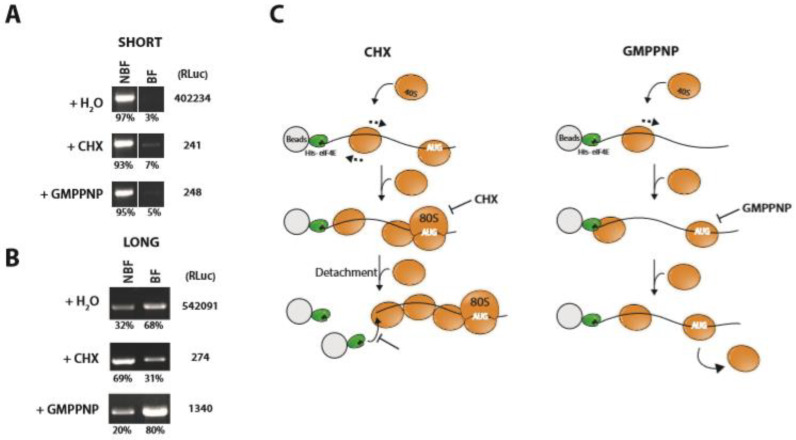
Solid phase expression from immobilized short and long TLs: impact of the drugs cycloheximide and GMPPNP. (**A**) Capped RLuc reporter transcripts with a short 5′ TL (8 nt) were immobilized via ^HIS^eIF4E as outlined in Figure 2. Beads were used to program cell extracts containing either H_2_O, cycloheximide to trap the 80S or GMPPNP to trap the 43S. RNA in the BF and NBF fractions was detected by RT-PCR. The percentage of the total in each fraction are indicated as is the measured RLuc reporter values. (**B**) Capped RLuc reporter transcripts with a long Kozak 5′ TL (50 nt) were immobilized via ^HIS^eIF4E as outlined in Figure 2. Beads were used to program cell extracts containing either H2O, cycloheximide to trap the 80S or GMPPNP to trap the 43S. RNA in the BF and NBF fractions was detected by RT-PCR. The percentage of the total in each fraction are indicated as is the measured RLuc reporter values. (**C**) Schematic models for the effect of the different drugs on the immobilized 5′ TL long reporter. Loading of the PIC onto the mRNA is probably the rate-limiting step during the initiation phase. Left hand panel: The 80S ribosome is frozen or clamped on a start codon due to the presence of CHX and this causes queuing of scanning ribosomes upstream. The arrows around the 40S indicates bidirectional scanning (see text). This ultimately impedes eIF4E recycling as the last recruited PIC will sterically block accessibility to the 5′ cap. The eIF4E carrying a black circle depicts the immobilized protein on the bead that will compete for the cap with the untagged protein in the cell extract. Ultimately, queuing will drive the transcript from the bead. Right hand panel: the presence of GMPPNP prevents 60S recruited but does not appear to induce the queuing effect observed with CHX. It is possible that under these conditions the PIC paused over the AUG codon is disrupted or displaced by the upstream scanning ribosome.

**Table 1 ijms-23-13369-t001:** Leakiness of the first AUG initiation codon as measured by the relative ratios of the AUG^ELK^ and iAUG^a/b/c^ bands on the immunoblots quantitated using ImageLab (i.e., the percentage of proteins originating from the downstream AUG codons: the higher the score, the more “leaky” is the first AUG).

	LEAKINESS			
	VacT7/HeLa	mRNA/HEK293T	RRL	HeLa Extract
G	95	95	36	68
GGG	30	30	22	43
Rdm	85	85	22	32
TISU	34	34	0	19
Kozak	5	5	0	6

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
