# Peer review of "Preinitiation Complex Loading onto mRNAs with Long versus Short 5′ TLs"

_ijms, 2022, doi:10.3390/ijms232113369_

Round 1
Reviewer 1 Report
Despite the decades of research, mechanics of scanning literally remains terra incognita. Specifically, one of our blind spots is translation initiation on mRNAs with very short 5’ UTRs. While there can be no doubt that ribosomes start to inspect mRNA right from its very 5’ terminal nucleotide, there can likewise be no doubt in admitting lack of understanding why cap-binding protein eIF4E does not interfere with this. I feel, therefore, saying any research on this topic is welcome is a huge underestimation. However, the work suffers from serious flaws, which I outline below. In general, the manuscript consists of two parts that do not combine in one piece. The first one hardly shows new data, while the second one – quite intriguing! – is just one fact.
First, several technical issues make the data ambiguous.
1. Using vaccinia virus for capping does not seem to be a proper choice, especially in the absence of an adequate RNA integrity analysis. A huge fraction of the T7-produced mRNA remains uncapped and this depends (among other factors) on the 5’-proximal RNA sequence (Fuerst and Moss 1989). This can (at least in part) explain the discrepancy in leaky scanning potential. (A) Non-capped mRNA may be more prone to leaky scanning through 5’ proximal AUGs. I’m not sure if I have ever seen any relevant data, except maybe the TISU paper where they failed to detect a sufficient amount or the reporter (Elfakess et al. 2011). (B) In cells, the absence on m7G-cap might lead to a 5’-to-3’ degradation of the T7-produced mRNAs which would result in an elimination of the 5’-proximal AUG(ELK) and, therefore, disappearance of the longer translation product. This issue can be resolved by comparing translation of m7G- and A-capped in vitro transcribed mRNAs. Otherwise, RNA expression levels and capping efficiencies should be addressed. Likewise, the notion above may explain the difference between the G and GGG reporters. GGG is a better template to T7 RNAP, thus it will produce more mRNA that will inevitably has a lesser fraction of the capped molecules, which will result in the apparent leakiness.
2. The lysates were collected 24h post mRNA transfections. The transfected mRNA likely ceases to be by this time. In their previous work, the authors showed quite short lifetime of their reporter peptide (Legrand et al. 2014) and thus it is totally unclear, do the reporter remnants seen on the blot reflect any steady-state pattern of translation?
3. In vitro translation systems are such a powerful tool that it is possible to unintentionally obtain literally any (un)desired data. Cap-dependence, context recognition stringency, translation efficiency, inhibition by cap-analogs or 4E-binding proteins, etc are all affected by ionic conditions. Therefore, any lysate should be tested and ionic conditions should be optimized with the reference to in vivo data.
4. The high cap-dependence of the bicitronic reporters (Supplementary Figure 2) is actually surprising. There are two reports that showed stimulation of a first cistron by EMCV or FMDV IRESs (Terenin et al. 2013; Jünemann et al. 2007). I feel the reason for this discrepancy may be hidden in the RNA production design. First, uncapped mRNA may be prone to degradation as mentioned above. Therefore, it would be wise to compare A-capped vs. m7G-capped mRNAs. Second, templates for transcription were linearized by XbaI, which is nearly the worst option possible. The enzyme cleaves DNA just 2 nt downstream of the Fluc stop codon and very short 3’ UTR impair translation (Tanguay and Gallie 1996a, 1996b; Lashkevich et al. 2020). This most likely does not affect anything, but should be discussed or better redone properly.
The idea that it is possible to attach biotin to mRNA 5’ terminus puts an idea in my head. Could it be feasible to create a chimaera of SBP and the eIF4G-binding peptide from eIF4E and a tag for mobilization and purification? The SBP-biotin interaction with Kd = 2.5 nM is way stronger than cap-eIF4E interaction and can be disrupted by an addition of biotin. An interesting tool it might be. Alternatively, cap-biotin (Bednarek et al. 2018) or eIF4E mutant with higher affinity to m7G-cap (Friedland et al. 2005) are definitely exploitable. Will this mutant inhibit translation of “long” and “short” mRNAs equally?
Also, it doesn’t seem correct to call ˜50 nt leader long. It’s way shorter than the average for mammals. Yet, let’s leave the terminology aside. The real question is is the long leader long enough? For the shorter TLs one can be sure that only one ribosome can sit there. “The long” TL is 50 nt long and its 15 nt are occupied by the initiating ribosome. Are these 35 nt sufficient to accommodate another scanning ribosome? My gut feeling is yes, but as scanning complexes occupy up to 80 nt of mRNA (Bohlen et al. 2020; Archer et al. 2016), certain degree of uncertainty remains.
As for the difference between the CHX- and GDPNP-stalled complexes. Indeed, lack of GTP hydrolysis may force 48S to resume scanning in an artificial system (Terenin et al. 2016), which probably can be promoted by upstream scanning ribosomes “pushing” the 48S complex. However, the difference may reflect distinct composition of 48S and 80S complexes. For example, if the m7G-eIF4F-eIF3-40S link is reestablished right after the 40S accommodates the mRNA and then remains intact until 80S forms, you would see exactly what you see, wouldn’t you?
MINOR POINTS
eIF4E was reported not to crosslink to m7G cap in 48S complexes (Kumar et al. 2016). This fact should be mentioned and discussed.
LANE 71. I would avoid saying that TISU translation is eIF4A-independent. It’s somewhat refractory to the eIF4A dominant-negative mutant action indeed (Elfakess et al. 2011), but when they reconstructed 48S complexes from purified components they for some reason didn’t show what happens in the absence of eIF4A (Sinvani et al. 2015).
LANE 99. I’d suggest avoiding the use of the verb “to promote”. Short leaders do not promote leaky scanning, they are more susceptible to leakiness than longer ones.
LANE 237. Mammalian and yeast translation initiation mechanisms fundamentally differ in certain aspects. For example, while eIF3 is required for 40S-mRNA binding in mammals, it isn’t in yeast. Thus, the fact that eIF4E-43S link is disrupted in yeast doesn’t necessarily mean that the same is true for mammals, yet it’s what is anticipated indeed.
LANE 376. There is likely no re-binding of eIF4E in the case of short 5’TLs. Since the 48S most likely isn’t linked to the m7G-cap, no cap-eIF4F interaction will occur until another 40S loads to the mRNA.
Supplementary Figure 2. Please avoid showing SEMs. SD!
To sum up. I find this direction of investigation very promising. However, to be suitable for publishing, the manuscript should be upgraded.
1. Vaccinia capping system should be replaced with in vitro transcription of m7G- and A- capped reporters.
2. Plasmids for transcription should be linearized by HpaI or (better) a polyA-tail should be introduced. At any rate, presence or absence of polyA-tail should be mentioned.
3. Analysis of the reporter translation in cells should be performed way earlier than 24 h posttransfection.
4. In vitro translation lysate should be shown to successfully recapitulate cap-dependence observed in cells and adequate inhibition by a 4EBP. This is crucial.
5. Solid-phase translation should be repeated with a couple of shorter 5’TLs, e.g., 20 and 30 nt. This will demonstrate at which distance from the 5’ end eIF4E-cap re-binding occurs and if it coincides with the next loading event. On the other hand, using a longer leader (e.g., 100 nt) ensures not only loading of a PIC (apart from 48S/80S residing on the start codon), but it also ensures the loaded complex does scan the TL after accomplishing the initial binding.
Archer SK, Shirokikh NE, Beilharz TH, Preiss T. 2016. Dynamics of ribosome scanning and recycling revealed by translation complex profiling. Nature 535: 570–574. doi:10.1038/nature18647.
Bednarek S, Madan V, Sikorski PJ, Bartenschlager R, Kowalska J, Jemielity J. 2018. mRNAs biotinylated within the 5′ cap and protected against decapping: new tools to capture RNA–protein complexes. Philosophical Transactions Royal Soc B Biological Sci 373: 20180167. doi:10.1098/rstb.2018.0167.
Bohlen J, Fenzl K, Kramer G, Bukau B, Teleman AA. 2020. Selective 40S Footprinting Reveals Cap-Tethered Ribosome Scanning in Human Cells. Mol Cell 79: 561-574.e5. doi:10.1016/j.molcel.2020.06.005.
Elfakess R, Sinvani H, Haimov O, Svitkin Y, Sonenberg N, Dikstein R. 2011. Unique translation initiation of mRNAs-containing TISU element. Nucleic Acids Res 39: 7598–7609. doi:10.1093/nar/gkr484.
Friedland DE, Wooten WNB, LaVoy JE, Hagedorn CH, Goss DJ. 2005. A mutant of eukaryotic protein synthesis initiation factor eIF4E(K119A) has an increased binding affinity for both m7G cap analogues and eIF4G peptides. Biochemistry-us 44: 4546–50. doi:10.1021/bi047645m.
Fuerst TR, Moss B. 1989. Structure and stability of mRNA synthesized by vaccinia virus-encoded bacteriophage T7 RNA polymerase in mammalian cells. Importance of the 5’ untranslated leader. J Mol Biol206: 333–348. doi:10.1016/0022-2836(89)90483-x.
Jünemann C, Song Y, Bassili G, Goergen D, Henke J, Niepmann M. 2007. Picornavirus internal ribosome entry site elements can stimulate translation of upstream genes. J Biol Chem 282: 132–141. doi:10.1074/jbc.m608750200.
Kumar P, Hellen CUT, Pestova TV. 2016. Toward the mechanism of eIF4F-mediated ribosomal attachment to mammalian capped mRNAs. Genes Dev 30: 1573–1588. doi:10.1101/gad.282418.116.
Lashkevich KA, Shlyk VI, Kushchenko AS, Gladyshev VN, Alkalaeva EZ, Dmitriev SE. 2020. CTELS: A Cell-Free System for the Analysis of Translation Termination Rate. Biomol 10: 911. doi:10.3390/biom10060911.
Legrand N, Araud T, Conne B, Kuijpers O, Jaquier-Gubler P, Curran J. 2014. An AUG Codon Conserved for Protein Function Rather than Translational Initiation: The Story of the Protein sElk1. Plos One 9: e102890. doi:10.1371/journal.pone.0102890.
Sinvani H, Haimov O, Svitkin Y, Sonenberg N, Tamarkin-Ben-Harush A, Viollet B, Dikstein R. 2015. Translational tolerance of mitochondrial genes to metabolic energy stress involves TISU and eIF1-eIF4GI cooperation in start codon selection. Cell Metab 21: 479–492. doi:10.1016/j.cmet.2015.02.010.
Tanguay RL, Gallie DR. 1996a. The effect of the length of the 3′‐untranslated region on expression in plants. Febs Lett 394: 285–288. doi:10.1016/0014-5793(96)00970-2.
Tanguay RL, Gallie DR. 1996b. Translational efficiency is regulated by the length of the 3’ untranslated region. Mol Cell Biol 16: 146–56. doi:10.1128/mcb.16.1.146.
Terenin IM, Akulich KA, Andreev DE, Polyanskaya SA, Shatsky IN, Dmitriev SE. 2016. Sliding of a 43S ribosomal complex from the recognized AUG codon triggered by a delay in eIF2-bound GTP hydrolysis. Nucleic Acids Res 44: 1882–1893. doi:10.1093/nar/gkv1514.
Terenin IM, Andreev DE, Dmitriev SE, Shatsky IN. 2013. A novel mechanism of eukaryotic translation initiation that is neither m7G-cap-, nor IRES-dependent. Nucleic Acids Res 41: 1807–1816. doi:10.1093/nar/gks1282.
Author Response
At the outset, I would like to thank both reviewers for their comments and most specially reviewer #1 for his clinical analysis of our manuscript. Many of his comments are pertinent and there realisation would most certainly add to the study. Altering 5’ TL length, as proposed, could reveal further insights into what is happening in our PIC loading assays. This represents a more extensive study and with the lab closing shorty this is for us “non-realisable”. One would hope that our results would stimulate others to complete the study. However, we would like the reader of this short manuscript to take away two messages. Firstly, that one can observe initiation at 5’ proximal start sites in multiple in-vivo and in-vitro systems independent of a “TISU context” (this is frequently cited in literature when discussing initiation on short 5’ TLs: e.g. Leppek et al., 2018 DOI: 10.1038/nrm.2017.103 or Kwan and Thompson, 2019 doi: 10.1101/cshperspect.a032672, to name a few). This has important implications when bioinformaticians seek to predict the mammalian proteome. Secondly, that the solid phase assay reveals differences in the behaviour of the initially immobilised mRNA that depends on TL length. Concerning the specific points raised at the end of the review, we have the following response:
- Vaccinia capping system should be replaced with in vitro transcription of m7G- and A- capped reporters.
We tested both the vaccinia capping system and in-vitro transcription using the m7G dinucleotide. We observed that the latter system gave lower RNA yields, presumably because one has to reduce the GTP concentration to promote transcription initiation with the cap dinucleotide, and capping efficiency was never as good as that observed with the capping enzyme. It is true that the Moss lab did report that sequences close to the cap could influence capping efficiency but these formed RNA structures. The sequences 5’ in all our constructs contain no evident RNA structure and all seem to cap with equal efficiency (>80%) as judged by binding to immobilised eIF4E (new Figure S2). This is consistent with other studies (Beverly et al., 2016, DOI: 10.1007/s00216-016-9605-x and Vlatkovic et al., 2022, doi.org/10.3390/pharmaceutics14020328). Our results also indicate that the in-vitro vaccinia capping system is more efficient than in vitro priming with m7GpppN dinucleotide and produces much higher yields of RNA. Using this approach, it is more straightforward to compare the uncapped initial transcript to the same transcript after cap addition (rather than using A-capped transcripts: although we accept this would probably be the better control).
- Plasmids for transcription should be linearized by HpaI or (better) a polyA-tail should be introduced. At any rate, presence or absence of polyA-tail should be mentioned.
This was an oversight in the previous version. All transcripts were polyA tailed as now demonstrated in the new Figure S2.
- Analysis of the reporter translation in cells should be performed way earlier than 24 h posttransfection.
It was standard procedure to harvest at 24 hrs in both the vaccinia-T7 transient assay with plasmid DNA (Figure 1D) and in the transient mRNA transfections (Figure 1G). With the latter, it is likely that mRNA stability (which would be similar for all transcript variants) limits de-novo synthesis to the initial ~6hrs post-transfection. Since both protein products have similar decay rates, the levels observed on the immunoblots are still reflective of the synthesis rate during this earlier phase. Harvesting earlier would have increased the signals but would not have impacted on the relative abundance of both products.
- In vitro translation lysate should be shown to successfully recapitulate cap-dependence observed in cells and adequate inhibition by a 4EBP. This is crucial.
I think no one who works with RRL will be surprised by the lack of cap dependence. Concerning our own mammalian cell extracts, we followed the protocol developed in the Shatsky group. They have already published that these cell extracts respond both to cap analogues and 4EBP1 (Andreev et al., 2009, DOI: 10.1093/nar/gkp665).
- Solid-phase translation should be repeated with a couple of shorter 5’TLs, e.g., 20 and 30 nt. This will demonstrate at which distance from the 5’ end eIF4E-cap re-binding occurs and if it coincides with the next loading event. On the other hand, using a longer leader (e.g., 100 nt) ensures not only loading of a PIC (apart from 48S/80S residing on the start codon), but it also ensures the loaded complex does scan the TL after accomplishing the initial binding.
We agree completely with the experiments proposed by the reviewer. Hopefully, upon reading of our short manuscript some other group will be inspired to apply this approach to probe 5’ TL length in greater detail.
Finally, we have re-organised the figures and the text (highlighted in red) in the hope that it is now more readily. We hope that you can consider this revised manuscript suitable for publication.
Reviewer 2 Report
This brief report investigates the translation initiation of short and long transcript leader in in vivo and in vitro contexts. The authors demonstrate that translation from a short 5’ transcript leader is cap-dependent and conclude that long and short transcript leaders have mechanistic differences in their mode of 43S loading. It would have been good if the conclusion section had succinctly contextualized the above conclusions with reference to the various models of 43S recruitment.
The experiments are well controlled and described in the material and methods section.
Minor comments to address:
1. Define RRL when first used in the text.
2. The MYC tag is not shown in Figure 1B.
3. Figure 2. Legend does not include labels for elements E and F.
4. Figure 3. The legend is confusingly written and needs to be rewritten more clearly.
Author Response
I hope he will find that his corrections have been made in the revised manuscript.
Round 2
Reviewer 1 Report
Sad to hear the lab is closing. All I can wish then is good luck in future, peaceful sky above, and please expand the text a bit to highlight limitations of the study and possible future directions.
Author Response
I have now added a new final paragraph that deals with most of the issues raised by the reviewer. Once again, I thank him/her for their very constructive and understanding review. I hope they find it now suitable for publication.
